# Alpine treeline ecotone stasis in the face of recent climate change and disturbance by fire

**Aviya Naccarella**[1,2]*, **John W. Morgan**[1,2], **Seraphina C. Cutler**[1], **Susanna E. Venn**[1,2,3]

1 Research Centre for Applied Alpine Ecology, La Trobe University, Bundoora Victoria, Australia,
2 Department of Ecology, Environment and Evolution, La Trobe University, Bundoora Victoria, Australia,
3 Centre for Integrative Ecology, School of Life and Environmental Sciences, Deakin University, Burwood, Victoria, Australia

* aviyanaccarella@gmail.com

**Data Availability Statement:** The data that support the findings of this study is openly available in Dataverse at https://doi.org/10.7910/DVN/I5MNND.

## Abstract

How species respond to climate change will depend on biological characteristics, species physiological limits, traits (such as dispersal), and interactions with disturbance. We examine multi-decadal shifts in the distribution of trees at the alpine treeline in response to regional warming and repeated disturbance by fire in the Victorian Alps, south-east Australia. Alpine treelines are composed of *Eucalyptus pauciflora* subsp. *niphophila* (Snow Gum, Myrtaceae) species. The location and basal girth of all trees and saplings were recorded across treelines at four mountains in 2002 and 2018. We quantify changes in treeline position (sapling recruitment above treeline) over time in relation to warming and disturbance by fire, and examine changes in stand structure below treeline (stand density, size class analyses). Short-distance advance of the treeline occurred between 2002 and 2018, but was largely restricted to areas that were unburned during this period. No saplings were seen above treeline after two fires, despite evidence that saplings were common pre-fire. Below treeline, subalpine woodland stands were largely resilient to fire; trees resprouted from lignotubers. However, small trees were reduced in number in woodlands when burned twice within a decade. Population dynamics at the alpine treeline were responsive to recent climate change, but other factors (e.g. disturbance) are crucial to understand recruitment trends. Establishment of saplings above treeline was largely restricted to unburned areas. These results indicate fire is a strong demographic filter on treeline dynamics; there is a clear need to frame alpine treeline establishment processes beyond just being a response to climate warming. Long lag periods in treeline change may be expected where recurrent disturbance is a feature of the landscape.

## Introduction

Alpine treelines are conspicuous and sharp transition zones, defining the upper elevational limit of tree growth [1, 2]. Limits to tree growth across the globe are widely reported to occur as a consequence of low mean growing-season soil temperature [3], but other less well-studied factors (e.g. competition, herbivory, availability of safe sites, frost, disturbance) also influence the position of the treeline at local-scales [4–8].

**Funding:** The Research Centre for Applied Alpine Ecology provided financial support.

**Competing interests:** The authors have declared that no competing interests exist.

Rising global temperatures are predicted to improve conditions for tree growth above the current treeline, leading to an upslope advance of trees [6, 9, 10]. However, Harsch et al. [10] found in a meta-analysis of 166 sites that 47% of alpine and high latitudinal treelines had shown no sign of advance, despite regional warming. It is likely that other local and site-specific factors, typically occurring at small-spatial scales, currently override the influence of increased global temperature on tree establishment above treeline [7]. Factors that affect seedling germination and early establishment are likely very influential [5–8, 10, 11].

Disturbance can have a significant effect on treeline position, either as an agent of direct mortality of adult trees [12], and/or via promotion of seedling recruitment opportunities [13], particularly in serotinous species [2, 14]. Fire, as one agent of disturbance, can act as a positive catalyst for treeline boundary change [2], triggering seed release, reducing ground layer competition, and generating bare ground opportunities for a pulse of seedling establishment [1, 13, 15]. Alternatively, fire may have negative effects on treeline boundaries if short-return intervals of fire lead to an "interval squeeze" [16], whereby the disturbance frequency reduces stand regeneration opportunities and produces an 'immaturity risk' at the population level [17]. This vulnerability will be highest in seeder treeline species whereas resprouting species are more likely to persist *in situ* [16, 18–21]; seedling regeneration opportunities may, however, be limited where fire frequency exceeds the secondary juvenile period of such trees [22]. Hence, treelines composed of resprouter species could theoretically remain unchanged following repeated fire [18, 23], reinforcing the existence of a sharp, non-mobile boundary. Resprouting is, however, a population attribute rather than a species attribute *per se* [20]. Resprouting response to fire disturbance varies in relation to two key parameters: 1) regime, which includes fire-return interval [12, 18, 24] and, 2) life-history stage [24, 25]. In some species, resprouting ability increases with size to reach a maximum in adult stages while in other species, resprouting is common in juveniles but adults are unable to resprout [18, 25]. In others again, resprouting capacity can decrease with recurrent short-interval wildfire [26]. Hence, treeline dynamics are likely to be complex in fire-prone landscapes, and fire-treeline feedbacks will likely assume more prominence in coming decades as climate conditions increase the likelihood of fire in mountain ecosystems [7, 20, 27–29].

Studies that quantify multi-decadal stand structural change at alpine treelines in the face of climate change and disturbance are necessary to interpret the wide variety of treeline responses reported [7]. Here, we assess the stability of alpine treelines in response to regional climate warming and repeated fire—where treelines are composed of broad-leaved, evergreen, facultative resprouter trees—using a re-visitation approach in the alpine areas of south-eastern Australia. Specifically, we asked: what evidence is there for changes in treeline position in response to recent climate warming and how is this affected by fire disturbance? And, has fire affected woodland stand structure below the treeline via effects on tree mortality and/or recruitment?

We hypothesised that:

1. Treelines without fire disturbance (since the time of first survey, $Time_1$) will have upslope sapling recruitment (at $Time_2$) due to recent climate warming. The stand structure below treeline will remain stable (Fig 1A).

2. Facultative seeder treelines with a single fire disturbance (since the time of first survey, $Time_1$) will have a pulse of sapling recruitment after fire and most of this will occur near adult trees at / above treeline (at $Time_2$) due to limited dispersal. The stand structure below treeline will remain stable because of resprouting (Fig 1B).

3. Facultative seeder treelines with repeated fire disturbances will show evidence of an 'interval -squeeze' with no recruitment above treeline since $Time_1$ (Fig 1C). The stand structure

below treeline will change, but not directionally, as not all resprouting trees will survive repeated fire (Fig 1C). Treeline regression will occur at Time$_2$ if trees at the climatic limit of growth are most vulnerable to repeated fire (Fig 1D).

## Methods

### Study species

Alpine treelines in the Australian Alps are dominated by a single tree species, *Eucalyptus pauci-flora* subsp. *niphophila* (Snow Gum, Myrtaceae, (Maiden & Blakely) L.A.S.Johnson & Blaxell), hereafter *Eucalyptus pauciflora*. Treelines vary from ~1750 m to 1900 m asl in the Australian Alps, varying with latitude and aspect [15]. *Eucalyptus pauciflora* is a long-lived species (i.e. potentially living hundreds of years to millenia), possessing a lignotuber from which resprout-ing occurs following canopy damage and produces a multi-stemmed growth form [15, 23]. Barker [30] suggests that trees with >250 cm basal girth are >300 yrs of age.

*Eucalyptus pauciflora* is capable of regenerating by seed and resprouting from meristematic tissue in lignotubers [31]. Dispersal is short, largely confined to within a few widths of the can-opy [15]. It is a "niche persister" [18] due to its capacity to re-occupy sites after disturbance, a trait most likely derived from the severe climatic conditions which constrain seedling estab-lishment at high elevations [32–35]. The ability of *E. pauciflora* to resprout suggests that re-occupancy after disturbance by fire is likely to be high. Similarly, understorey vegetation (including many shrubs, graminoids and forbs) re-sprout or are able to germinate from the

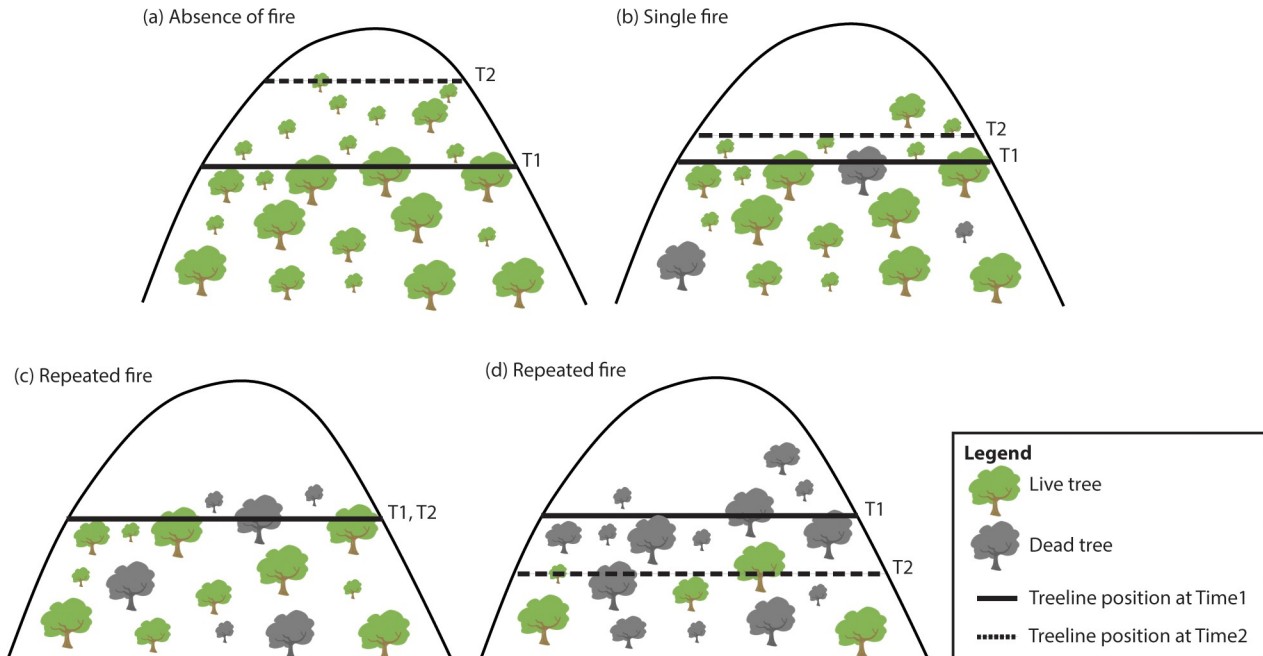

**Fig 1. Change in treeline position over time is hypothesised to depend on interactions with both climate and fire.** (a) In the absence of fire, seedlings will have established above treeline due to regional climate warming. Stand structure will remain largely unchanged between Time$_1$ and Time$_2$. (b) After a single fire, there will be a pulse of sapling recruitment which will occur near adult trees at/above the treeline due to limited dispersal. Stand structure will remain stable between Time$_1$ and Time$_2$ due to resprouting. (c) After two fires, treelines will show evidence of 'interval-squeeze' with no recruitment above treeline. Stand structure will change, but not directionally, as not all resprouting trees will have survived the second fire. (d) After two fires, treeline regression will occur at Time$_2$ if trees at the climatic limit of growth are most vulnerable to repeated fire and thus tree mortality is high. Based on conceptual model of vegetation change at the alpine treeline ecotone developed by Cansler et al. [12], but adapted for Australian *Eucalyptus pauciflora* treelines.

soil seed bank post-fire, and thus the whole system recovers well from infrequent fire [36]. However, *E. pauciflora* resprouting capacity is dependent on lignotuber survival, which has been shown to decline with multiple fires [35], particularly when the fire intervals are short [20]. The capacity to resprout is also thought to be dependent on fire severity, plant vigor, size and bark thickness [31, 37]. Theoretically, *E. pauciflora* is capable of resprouting within 6 months of germination when the lignotuber is first formed [37,38]. However, seedling lignotubers are poorly protected in the soil and they may be vulnerable to fire [37,38]. Time to reproductive maturity of *E. pauciflora* saplings is not known; however, time to first reproduction after fire (the secondary juvenile period) occurs within 6 yrs (S. Cutler *unpublished data*).

## Study area

The study was conducted on alpine mountain peaks in the Victorian Alps, south-east Australia. Four mountains (Mt Hotham, Mt Feathertop, Mt McKay, The Twins) with obvious treeline ecotone boundaries were selected for study in 2002 to determine treeline position and woodland structure (Fig 2 and S1 Table). Grazing history varied across sites, with stock grazing removed across all sites since 2003 (S1 Table). These mountains were then variably exposed to fire (S1 Table). In 2018, we re-surveyed the same localities on all four mountains to quantify how treelines had changed in response to climate and fire since the first survey.

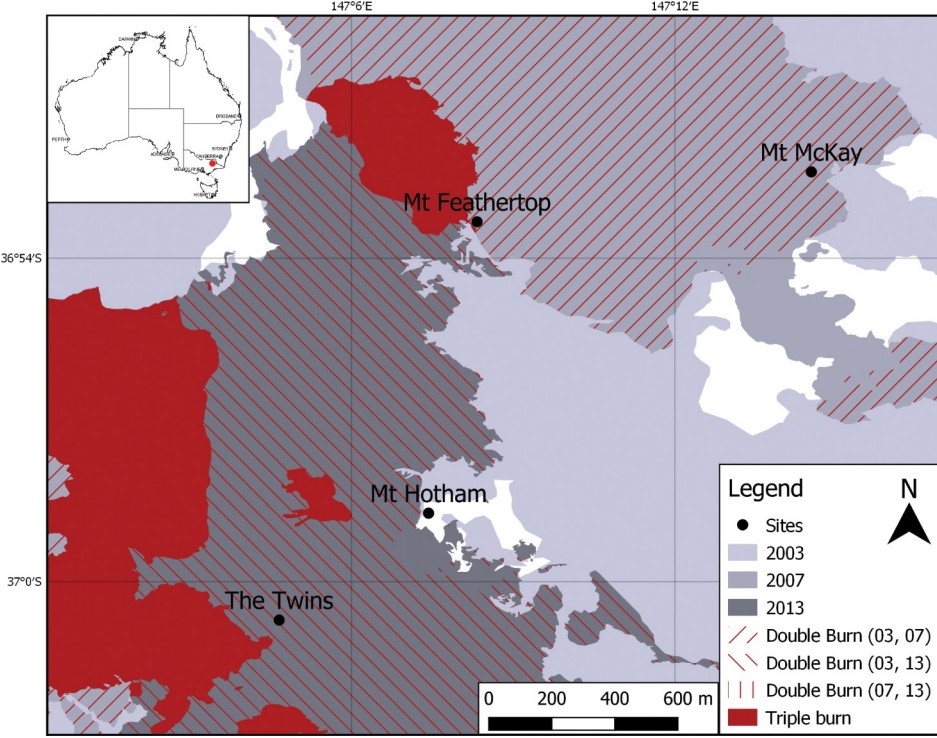

**Fig 2. Extent of 2003, 2007 and 2013 fires and site locations across the Victorian Alps, Australia.** Fire history at the transect scale varied from fire perimeters shown on this map. Mt Hotham was unburnt since 1939, Mt Feathertop and Mt McKay burnt once in 2003 and The Twins burnt twice in 2003 and 2013. Fire history data sourced from Victorian Government Department of Environment, Land, Water and Planning Spatial Datamart https://services.land.vic.gov. au/SpatialDatamart/dataSearchViewMetadata.html?anzlicId=ANZVI0803004741&extractionProviderId=1 under a CC BY license, with permission from Department of Environment, Land, Water and Planning, original copyright 2019.

The region is characterised by low temperatures, with an annual mean maximum temperature ranging between 8.0–9.4˚C and mean minimum temperature ranging between 1.9–2.6˚C (Mount Hotham meteorological station, Bureau of Meteorology, www.bom.gov.au). Annual mean precipitation is 1274–1454 mm, with much of it falling as winter snow. Frost frequency varies inter-annually. There has been an increase in mean annual minimum and maximum temperature of ~0.4˚C in the Victorian Alps since 1992. Temperatures are predicted to continue to warm by 0.6–2.9˚C by 2050 [39]. Extreme climate events (i.e. minimum temperature, maximum temperature, mean annual rainfall, maximum annual snow depth) have been more frequent since 2005[40]. Soils are largely of the alpine humus type and typically shallow near and above treelines (e.g. mean 8–32 cm deep; S. Cutler *unpublished data*).

Fires have historically been a relatively rare occurrence in the alpine regions of Australia, with a fire return interval of ~50–100 years [36, 41, 42], with all sites likely to have been burnt in 1939 [43]. Between 2000 to the time of measurement in 2018, much of the Victorian Alps has been burnt to varying degrees by three large-scale fires within a decade [44]. At our study sites, since 2001, fire had occurred 0, 1 or 2 times (Fig 2 and S1 Table), stimulating discussion about the effects of fire on alpine treeline dynamics and the resilience of treeline populations in the future, given fire frequency and severity is predicted to increase with climate change [35, 36, 42, 45]. This increase in fire frequency in the region is continuing post our repeat surveys, with large areas of the Australian Alps, including the Victorian Alps, burning during the 2019–2020 summer period.

## Field methods

In 2002, the position of the alpine treeline was determined at each mountain (corresponding with the upper extent of large adult trees (>3 m height and >25 cm basal girth)) [46] and 19 belt transects were established to sample stand structure below the treeline. Transect end-points were permanently marked. Belt transects were 5 m wide and extended for a distance of 40 m downslope of the treeline into subalpine low open woodland. In each transect, all *E. pauciflora* were assigned an *x* and *y* coordinate and the basal girth was measured. From this, size class distributions were deduced and age approximated [47]. Height of all individuals was also recorded. The area above the treeline was searched and transects extended to encompass all outpost individuals of *E. pauciflora* (e.g. trees, saplings, seedlings). We define *E. pauciflora* life forms as follows: 'trees' are individuals that have a basal girth greater than 25 cm, 'saplings' are individuals with a basal girth between 25 cm and ~0.13 cm and are estimated to be greater than 1 year old, and 'seedlings' are very young plants that are likely to be less than 1 year old corresponding to a basal girth of less than ~0.13 cm [47].

To determine the stability of the alpine treeline and woodland stand structure in response to regional warming and disturbance by fire, we relocated the original transects in 2018 using a combination of GPS co-ordinates, original field notes and permanent pegs. In transects where the original pegs could not be found, these were relocated based on GPS co-ordinates and transect maps from 2002. As such, any change over time represents local-scale changes in treeline dynamics and woodland structure as opposed to change in individual trees. Observed changes in treeline stability and woodland stand structure can be attributed to the effects of fires as windthrow or avalanches are uncommon, and Victorian snow gum woodlands are not yet affected by the *Phoracantha* beetle as in Kosciuszko National Park, Australia. A total of 19 transects were re-surveyed. Fire history, since 2002, was quantified at the transect scale by looking for evidence of burn scars and stem resprout cohorts; trees exposed to one fire have one live and one dead stem cohort, whereas trees exposed to two fires typically had two dead stem cohorts, and a smaller live stem cohort in 2018. Fire history at the transect scale varied from Victorian Government Department of Environment, Land, Water and Planning spatial

data (Fig 2). Of the 19 transects, five were unburnt since 1939 (Mt Hotham), eight burnt once in 2003 (Mt Feathertop, Mt McKay) and six burnt twice between 2003 and 2013 (The Twins). The location of trees were mapped as *x* and *y* coordinates. Basal girth, height and condition (live, dead) of trees was recorded.

## Data analysis

The year of establishment of individuals located above treeline was calculated from basal girth according to the methods of Rumpff et al. [47]. A Fisher exact test was used, as expected values were <5, to determine differences in the number of saplings (<25 cm basal girth) located above treeline between survey periods. Where transformation did not improve the distribution of the data non-parametric tests were used. A Mann-Whitney U test was used to determine differences in the distance of individuals above treeline between survey periods. A Mann-Whitney U test was used to determine differences in the height of saplings (<25 cm basal girth) above treeline between survey periods. A Kruskal-Wallis rank sum test was used to determine any relationship between basal girth and mortality (live or dead).

To examine changes in woodland stand structure over time, size-class distributions (SCD) were used based on a model presented by Condit et al. [48]. Individuals were grouped into 17 basal diameter classes (size class, in mm), based on the structural demography of *E. pauciflora* woodland elsewhere [23]. The number of live individuals was counted per size class. To accommodate uneven size class width, the number of individuals ($N_i$) was divided by the width of the size class (Eq 1).

$$n_i = 100(dbh_i + 1 - dbh_i) \tag{1}$$

This gives the abundance per size class ($n_i$). The midpoint of each size class and abundance was natural log (ln) transformed and a regression calculated for each site in each survey period. The slope (*z*) of the regression was then used as an indicator of population structure. Outcomes of this approach can be interpreted broadly as:

*Type 1*: stands not recruitment-limited as there are large numbers of juveniles, saplings and young trees relative to older trees. These stands are typified as having a reverse j-curve size class distribution and slopes are high negative values (< -1.0)

*Type 2*: stands are moderately recruitment-limited, with small numbers of sapling and juvenile trees relative to older trees; slopes are flatter, with y-values falling between 0 and -1.0.

*Type 3*: stands are severely recruitment-limited, resulting in a recruitment bottleneck, with few to no observable trees in the smaller size classes relative to older trees. These have positive slopes (>0).

## Results

A total of 83 saplings (<25 cm basal girth) were located above treeline on the four mountains in 2018, compared to 119 in 2002. In both 2002 and 2018, the majority of saplings were within 5–10 m of the treeline, and were <57 cm in height, (Fig 3 and S1 Fig).

There was a significant increase in the distance of saplings (<25 cm girth) above treeline at $T_2$ (2002 median: 2.04 m, 2018 median: 7.23 m) (Fig 3; p<0.001). There was also a significant increase in the height of saplings (<25 cm girth) located above treeline at $T_2$ (2002 median: 0.15 m, 2018 median: 0.2 m) (Fig 3; p<0.001).

Establishment above treeline in relation to fire history was varied (Fig 3). There was a significant increase in saplings above treeline at Mt Hotham (unburnt), and recruitment

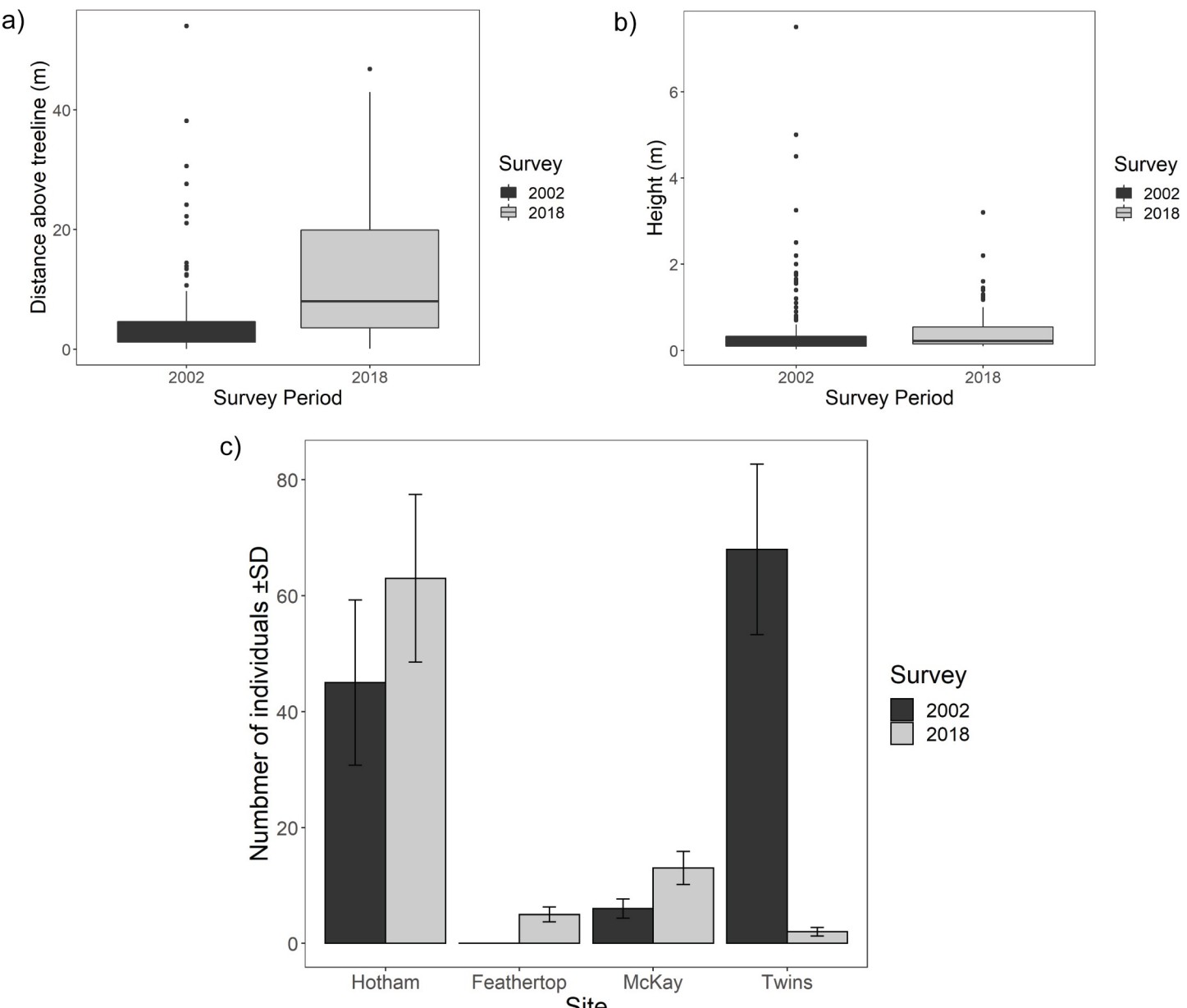

**Fig 3.** a) Distance above treeline of *Eucalyptus pauciflora* saplings (<25 cm girth) located above treeline in 2002 and 2018 surveys. b) Height of *Eucalyptus pauciflora* saplings (<25 cm girth) located above treeline in 2002 and 2018 surveys. c) Total number of *Eucalyptus pauciflora* saplings (<25 cm girth) above treeline (±standard deviation) in 2002 and 2018 surveys (Hotham n = 5; Mount Feathertop n = 4; McKay n = 4; The Twins n = 6).

continued between survey periods at the Mt Feathertop and Mt McKay (both single burn) sites (Fig 3; p<0.001). There was a significant decline in sapling number at The Twins between sampling times; this treeline was burnt twice between 2002 and 2018 (Fig 3; p = 0.008, and S1 Fig).

Estimates of the age of individuals above treeline in 2002 and 2018 indicate that most had established after 1995 and then again after 2012 (Fig 4). Few individuals (saplings and adults) above treeline in 2018 surveys had established before the 2002 survey (Fig 4), suggesting high turnover and thus mortality above treeline. However despite this there was low mortality across the whole transect (above and below treeline) as indicated by dead standing individuals

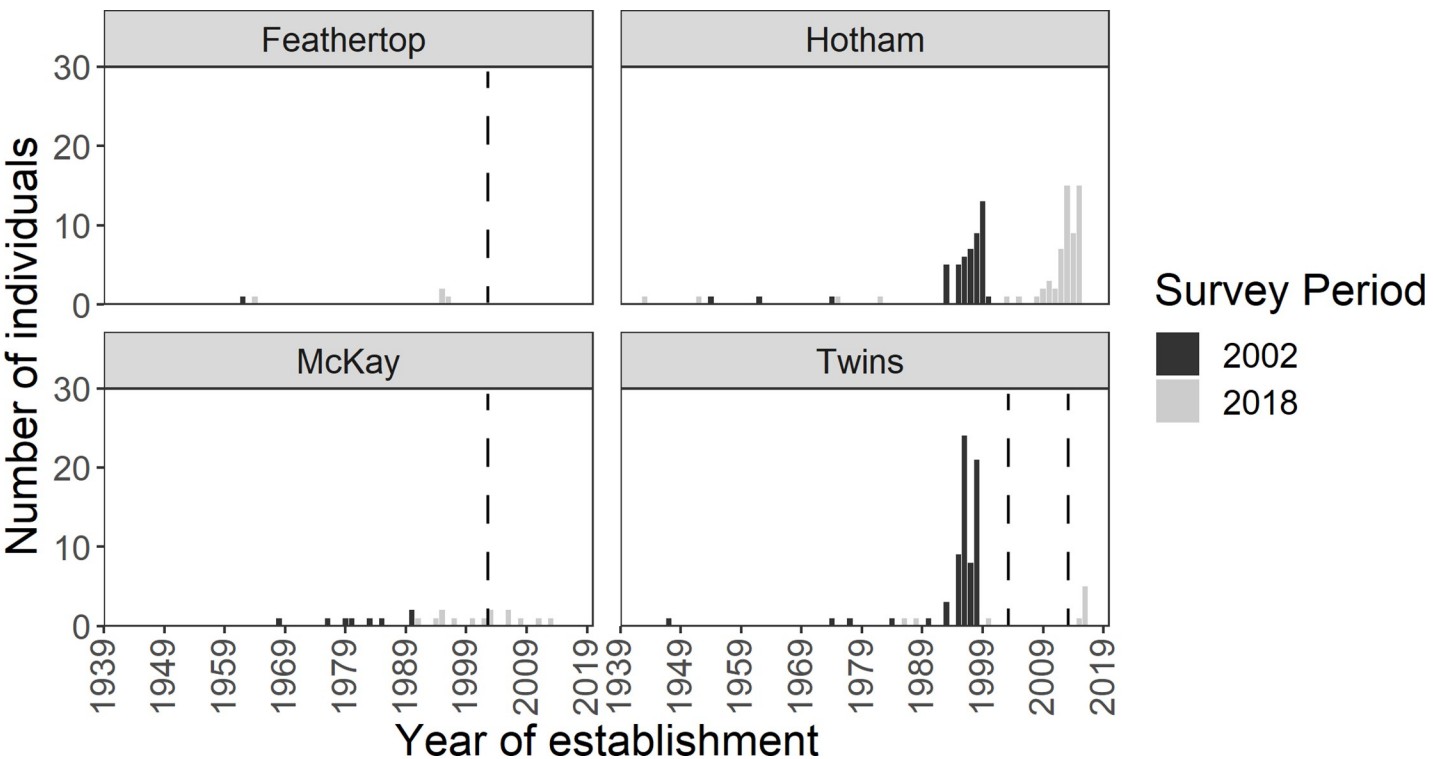

**Fig 4. Estimated year of establishment of *Eucalyptus pauciflora* individuals located above treeline in 2002 and 2018 surveys.** Vertical dashed lines indicate time of disturbance. Year of establishment calculated based on Rumpff et al. [46] model; trees with girth >115 cm (establishment pre-1938) were excluded.

found (0% unburnt sites, 8% single burn sites, 5% double burn sites). The basal girth of dead individuals was significantly higher than live individuals at single burn sites (p-value = 0.032). There was no significant difference between the basal girth of dead and alive individuals at the Twins twice-burnt site (p-value = 0.879).

Woodland stand structure below the treeline varied with burning history between 2002 and 2018 (Fig 5). At the unburnt mountain (Mt Hotham), SCD curves were more negative with time (2002: -0.798 (Type 2), $r^2$ = 0.86; 2018: -1.088 (Type 1), $r^2$ = 0.93) which indicates recruitment into small size-classes. By contrast, at the twice-burned mountain (The Twins), SCD curves flattened over time (2002: -1.043 (Type 1), $r^2$ = 0.92; 2018: -0.345 (Type 2), $r^2$ = 0.45), indicating a loss of individuals in smaller size-classes from the stand (Fig 4). At the once burned mountains, SCDs were stable (Mt McKay, 2002: -0.561, $r^2$ = 0.65; 2018: -0.509, $r^2$ = 0.52; both Type 2) or showed evidence of increasing (Mt Feathertop, 2002: -0.371 (Type 2), $r^2$ = 0.21; 2018: -1.1.132 (Type 1), $r^2$ = 0.70) (Fig 5 and S1 Fig).

## Discussion

How species respond to changing climates at local scales will depend on how edaphic and biological characteristics interact with species physiological limits, traits such as dispersal, and interactions with disturbance (type, frequency, severity). While climate is considered a key determinant of alpine treeline position [1], and hence treeline position is likely responsive to climate warming, disturbance may also play a critical but poorly understood role [12, 13, 49]. Using a revisitation approach at the alpine treeline in Australia where fire has been both extensive and recurrent during the last few decades [44], we found that a limited number of saplings of the alpine treeline species *E. pauciflora* recruited above the treeline, but this depended on

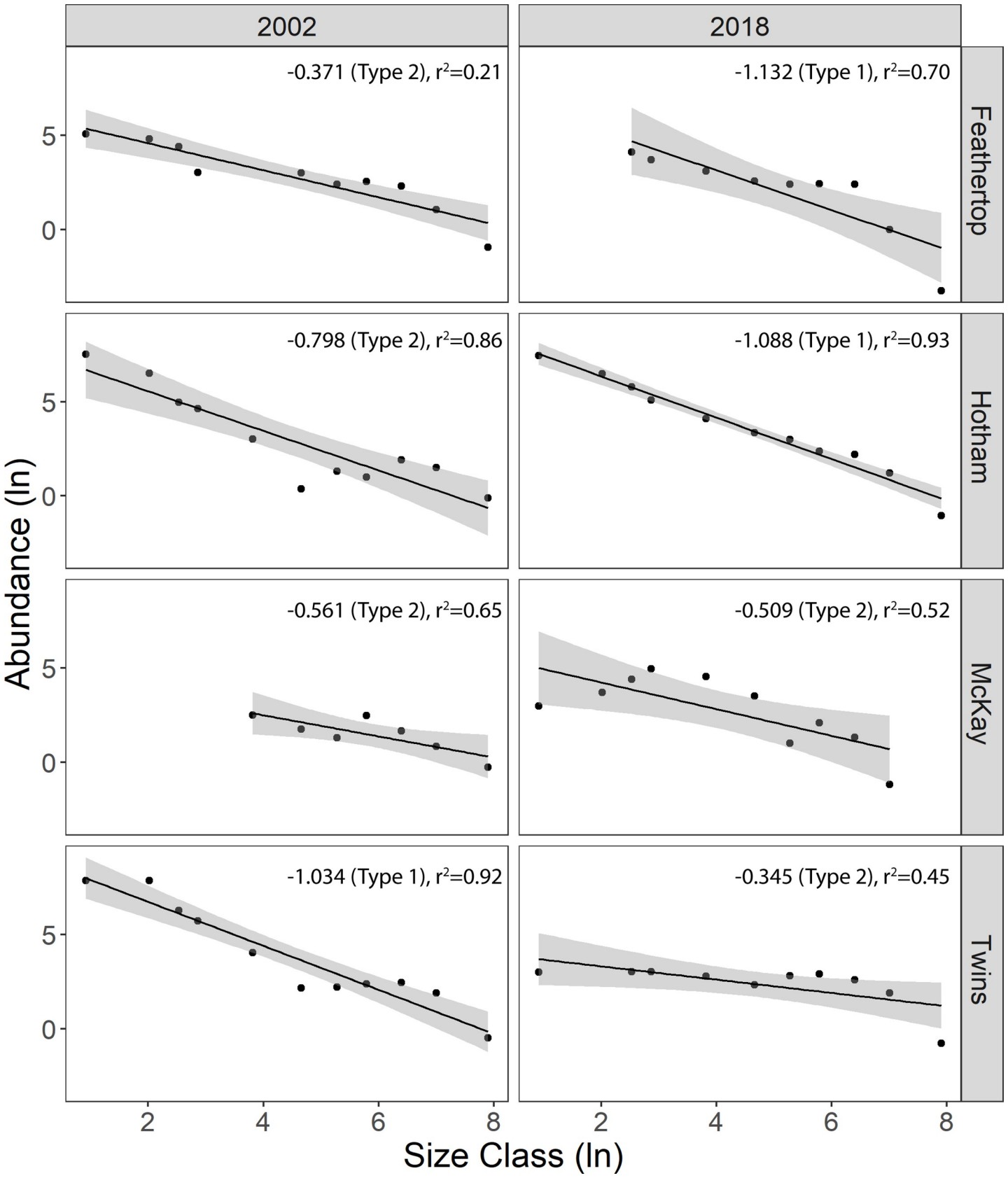

**Fig 5. Size class distribution of size class (natural log transformed) against abundance (individuals per size class corrected, natural log transformed) for** *Eucalyptus pauciflora* **individuals in 2002 (left panels) and 2018 (right panels) surveys.** Model is based on Condit et al. [48]. Population structure types; *Type 1*: stands not recruitment-limited as there are large numbers of juveniles, saplings and young trees relative to older trees. These stands are typified as having a reverse j-curve size class distribution and slopes are high negative values (< -1.0); *Type 2*: stands are moderately recruitment-limited, with small numbers of sapling and juvenile trees relative to older trees; slopes are flatter, with y-values falling between 0 and -1.0; *Type 3*: stands are severely recruitment-limited, resulting in a recruitment bottleneck, with few to no observable trees in the smaller size classes relative to older trees. These have positive slopes (>0).

the frequency of fire. In the absence of fire, short-distance (<10 m) regeneration was observed between 2002 and 2018. After two fires in this period, there was no sapling regeneration above treeline despite extensive recruitment in the period before fire (pre-2002). Fire frequency also affected the stand structure of subalpine woodlands. Few changes in structure occurred in the unburnt and once-burned forests; twice-burned stands, however, lost smaller-sized trees (<~20 cm basal girth) and this may be evidence of a fire 'interval squeeze' [16]. While changes in the environment, including climate warming, may allow saplings to grow above the current treeline, as has been observed in many alpine regions of the world [10], this capacity may depend on disturbance type and frequency. Hence, there is potential for long time lags in treeline movements in the Australian Alps because tree persistence and recruitment is affected by drivers other than just climate [7, 8, 50]. There is a clear need to frame alpine treeline establishment processes beyond just being a response to climate warming.

In the absence of fire, we hypothesised (Hypothesis 1) that ongoing climate warming (a ~0.4°C increase in mean annual temperature since 1992) would promote the upslope migration of the alpine treeline between our survey times (Time$_1$ 2002—Time$_2$ 2018). Trees at treeline are responsive to warming temperatures in many settings [6, 9,10] and hence, we expected to see establishment of *E. pauciflora*, in numbers that may be higher than observed prior to 2002, due to ongoing warming. We did observe ongoing sapling recruitment above treeline at Mt Hotham, the only mountain in our study that escaped wildfire between 2002 and 2018. Prior to 2002, saplings above established treelines were observed at Mt Hotham, establishing within 10 m of the treeline. This has continued, with most saplings still restricted to establishment near the treeline. We found a striking trend in the height (<~60 cm) and age (<~10 years) of individuals above treeline which was consistent across survey periods and sites. The age of individuals indicates high turnover and thus low longer-term survival above treeline. The height of individuals we observed suggests that a height threshold determined by freezing stress may exist [1,2]. Freezing stress is exacerbated immediately above the surrounding vegetation, corresponding to a shift from within the warm boundary layer, formed by maximum heat accumulation and retention near the soil surface, to above this layer where individuals become more closely coupled to atmospheric conditions [2, 15, 51,52]. Although mature *E. pauciflora* are relatively resistant to frost, individuals less than 30 cm tall can be easily killed by substantial shoot dieback [15]. Individuals thus require a period of frost-free damage to attain heights >1 m where frost occurrence and damage becomes less severe. This suggests that it may not be overall growing season temperature which limits sapling growth beyond this height, but the occurrence of extreme temperature events [6].

In unburned areas, limited upslope migration was observed. This may be an outcome of dispersal limitation, positive effects of nearby tree canopy on establishment (i.e. facilitation), or a combination of the two. Dispersal of *E. pauciflora* seed is limited, as it is for many eucalypts [53]. *Eucalyptus pauciflora* are generally short-statured at the treeline (i.e. <5 m tall), the seeds have no appendages to aid wind-dispersal, and modelled mean maximum dispersal distance (based on traits such as tree height and seed mass) is ~16 m (A. Naccarella *unpublished data*). Mortality of *E. pauciflora* germinants is likely highest during the first growing season (being dependent on summer soil water availability) and the first winter (being dependent on

the depth and duration of snow cover)[4]. Sapling survival is possible where the treeline canopy reduces photoinhibition [51], and may explain why most saplings are growing within 10 m of the edge of tree canopy. Additionally, canopy facilitation has been shown to reduce frost severity [51] and increase humidity buffering against drought [54–56], which could be resulting in a density-dependent positive feedback between canopy cover facilitation and seedling recruitment leading to restricted upslope migration [57, 58].

In the absence of disturbance by fire, woodland stand structure below treeline changed from Condit et al. [48] SCD curve Type 2 to Type 1, indicating that plants are being recruited into the population. SCD models indicate that woodland structure has remained stable at single burn sites like Mt McKay, with a higher proportion of saplings to mature individuals [48]. This suggests treeline populations are resilient to a single bushfire. SCD models at twice-burned treelines indicate a scarcity of saplings in 2018 (as indicated by lower slope values; Type 2), leading to a more even- aged stand [48]. Prior to being twice-burnt, saplings were prominent (Type 1), suggesting a shift in treeline dynamics over time with bushfire occurrence.

How *E. pauciflora* treeline responds to fire disturbance appears to be influenced by fire frequency. Once-burned areas show no evidence of pulsed seedling regeneration (counter to our Hypothesis 2), but there is little evidence of tree or sapling mortality, as quantified by size-class analyses conducted in 2002 and 2018. Twice-burned areas (within a decade), however, lose small-sized trees and saplings both above the treeline and within the subalpine woodland (supporting our Hypothesis 3). The occurrence of two fires within a decade is an unprecedented event in the Australian Alps [36] allowing the opportunity to explore the influence of short-interval fires on treeline dynamics. Although mature tree mortality was low, and as such there was no evidence of treeline recession, woodland structure had shifted substantially. The substantial reduction in saplings above- and below-treeline suggests younger and smaller individuals are more susceptible to fire. Despite *E. pauciflora* capacity to resprout within 6 months of germination, the lack of soil protection for the lignotuber could result in increased vulnerability, particularly during high-intensity fires [37, 38]. Additionally, the low stature of individuals above treeline likely increased their susceptibility to both (i) fire due to their close proximity to the ground layer and surface fuels [28] and (ii) frost due to the loss of groundlayer vegetation that may protect them from cold temperatures.

Understanding how disturbance, such as fire, impacts on treeline persistence and expansion is clearly crucial for interpreting treeline population dynamics [49,59]. Fires, particularly at short-intervals, have been found to cause treeline depression across a range of global treelines [28, 60–62]. Conversely, fires have also been found to accelerate the effect of rising temperatures through reducing competition with surrounding vegetation [15, 59, 49, 63]. Fire may assume an even greater importance in the Australian Alps in the future, with temperatures predicted to increase by 1.4–3.8°C over the next century for south-eastern Australia [64], increasing the frequency of high fire danger days by as much as 70% by 2050 [65,66]. Additionally, understanding how fire and climate may differentially or similarly affect lower elevation woodlands, including the lower montane *E. pauciflora* boundary, will inform how *E. pauciflora* ecotones could change in the future. Abrupt changes to the fire regime has already caused the landscape-wide loss of obligate seeder mountain forests in south-eastern Australia [44], suggesting there is the potential for shifts across many ecotone boundaries.

## Conclusion

The response of global alpine treelines to rising temperatures has been unpredictable and variable [10, 58, 67]. A combination of treeline advance, stability and retreat has been recorded, suggesting that temperature may not be the overriding control across many treelines. Hence,

upslope treeline movements lag behind climatic warming. Our results suggest that population dynamics at the alpine treeline in south-eastern Australia are responsive to climate warming, albeit at small spatial scales, and that other factors (e.g. disturbance regimes) are crucial for understanding recruitment. There is a clear need to frame alpine treeline establishment processes beyond just being a response to climate warming. Long lag periods in treeline change may be expected where recurrent disturbance is a feature of the landscape.

## Supporting information

**S1 Fig. A visual representation of *Eucalyptus pauciflora* individuals across representative transects expressing increases in individuals above treeline (Mount Hotham transect 1, W aspect), stability with similar numbers of seedlings above treeline (Mount Feathertop transect 3, W aspect) and decreases in seedlings above treeline (The Twins transect 2, N aspect).** X and Y axes indicate exact meter locations across the transect. Circle size indicates basal circumference in relative proportions to the X and Y axes. Grey = dead individuals. Black = live individuals. Treeline is represented by the red line at y = 40m, y<40 within the woodland, y>40 above treeline.
(TIF)

**S1 Table. Site characteristics (transect length, location (latitude and longitude presented in GDA94/ MGA zone 55, coordinates refer to treeline position centre), aspect, elevation, grazing history (Sourced from Lawrence [43], fire history since 2003 based on field assessments).**
(DOCX)

## Acknowledgments

We thank Martin Beilharz, Lisa Naccarella, Lucio Naccarella, Matthew Beattie, Honor Gillies and Paul McMorran for help in the field. Bev Lawrence (APEA) provided logistical support.

## Author Contributions

**Conceptualization:** Aviya Naccarella, John W. Morgan, Seraphina C. Cutler, Susanna E. Venn.

**Data curation:** Aviya Naccarella, John W. Morgan, Seraphina C. Cutler, Susanna E. Venn.

**Formal analysis:** Aviya Naccarella, John W. Morgan, Seraphina C. Cutler, Susanna E. Venn.

**Funding acquisition:** Aviya Naccarella, John W. Morgan, Seraphina C. Cutler, Susanna E. Venn.

**Investigation:** Aviya Naccarella, John W. Morgan, Seraphina C. Cutler, Susanna E. Venn.

**Methodology:** Aviya Naccarella, John W. Morgan, Seraphina C. Cutler, Susanna E. Venn.

**Project administration:** Aviya Naccarella, John W. Morgan, Seraphina C. Cutler, Susanna E. Venn.

**Resources:** Aviya Naccarella, John W. Morgan, Seraphina C. Cutler, Susanna E. Venn.

**Software:** Aviya Naccarella, John W. Morgan, Seraphina C. Cutler, Susanna E. Venn.

**Supervision:** Aviya Naccarella, John W. Morgan, Seraphina C. Cutler, Susanna E. Venn.

**Validation:** Aviya Naccarella, John W. Morgan, Seraphina C. Cutler, Susanna E. Venn.

**Visualization:** Aviya Naccarella, John W. Morgan, Seraphina C. Cutler, Susanna E. Venn.

**Writing – original draft:** Aviya Naccarella, John W. Morgan, Seraphina C. Cutler, Susanna E. Venn.

**Writing – review & editing:** Aviya Naccarella, John W. Morgan, Susanna E. Venn.

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
