## [Decision Letter · Decision Letter 0]

6 Jan 2020

PONE-D-19-29648

Alpine treeline ecotone stasis in the face of recent climate change and disturbance by fire

PLOS ONE

Dear Ms Naccarella,

Thank you for submitting your manuscript to PLOS ONE. After careful consideration, we feel that it has merit but does not fully meet PLOS ONE’s publication criteria as it currently stands. Therefore, we invite you to submit a revised version of the manuscript that addresses the points raised during the review process.

The two reviewers were positive, and both suggested minor revisions. Please check the reviewers' comments and include them in your work or explain why you did not apply the suggestions. I selected "major revisions", to give you more time  (if you need it) to complete the task.

We would appreciate receiving your revised manuscript by Feb 20 2020 11:59PM. To enhance the reproducibility of your results, we recommend that if applicable you deposit your laboratory protocols in protocols.io, where a protocol can be assigned its own identifier (DOI) such that it can be cited independently in the future. For instructions see: http://journals.plos.org/plosone/s/submission-guidelines#loc-laboratory-protocols

We look forward to receiving your revised manuscript.

Kind regards,

Sergio Rossi

Academic Editor

PLOS ONE

3. We note that you have stated that you will provide repository information for your data at acceptance. Should your manuscript be accepted for publication, we will hold it until you provide the relevant accession numbers or DOIs necessary to access your data. If you wish to make changes to your Data Availability statement, please describe these changes in your cover letter and we will update your Data Availability statement to reflect the information you provide

4. Thank you for stating the following in the Acknowledgments Section of your manuscript: "The Research Centre for Applied Alpine Ecology provided financial support"

 "The authors received no specific funding for this work."

5. We note that Figure S1 in your submission contain [map/satellite] images which may be copyrighted. All PLOS content is published under the Creative Commons Attribution License (CC BY 4.0), which means that the manuscript, images, and Supporting Information files will be freely available online, and any third party is permitted to access, download, copy, distribute, and use these materials in any way, even commercially, with proper attribution. For these reasons, we cannot publish previously copyrighted maps or satellite images created using proprietary data, such as Google software (Google Maps, Street View, and Earth). For more information, see our copyright guidelines: http://journals.plos.org/plosone/s/licenses-and-copyright.

a)    You may seek permission from the original copyright holder of Figure S1 to publish the content specifically under the CC BY 4.0 license.  

Reviewers' comments:

Reviewer's Responses to Questions

**Comments to the Author**

1. Is the manuscript technically sound, and do the data support the conclusions?

Reviewer #1: Yes

Reviewer #2: Yes

2. Has the statistical analysis been performed appropriately and rigorously? 

Reviewer #1: Yes

Reviewer #2: Yes

3. Have the authors made all data underlying the findings in their manuscript fully available?

Reviewer #1: Yes

Reviewer #2: Yes

4. Is the manuscript presented in an intelligible fashion and written in standard English?

Reviewer #1: Yes

Reviewer #2: Yes

5. Review Comments to the Author

Reviewer #1: Review Naccarella et al.

Alpine treeline ecotone stasis in the face of recent climate change and disturbance by fire

General comments:

Naccarella and colleagues revisited vegetation surveys from different locations at the alpine treeline in south-eastern Australia, to assess the impact of forest fires on upslope movement of treelines and forest stand structure. They could clearly show that in areas with an unusually high fire frequency, sapling establishment was reduced both within the subalpine forests and above treeline. They conclude that fire has the potential to impede the upslope movement of treelines in response to a warming climate.

I’ve greatly enjoyed reading this well-written and concise manuscript. The authors make a very convincing case for fire as an important secondary control of treeline dynamics. This is an important finding that could explain the lack of evidence of an upward shift of mountain forests in some parts of the world, and has implications for assessing future vegetation dynamics under climate change. Of course, it is a pity that the exact transects could not have been reproduced between sampling years. Nevertheless, the authors can show clear trends in response to fire on the local stand-scale. Even though I mostly agree with the interpretation and conclusions of the authors I have two remarks:

- To use the presence of seedlings/saplings above treeline to infer a response to warming temperatures is a bit tricky. As you state yourself, seedlings/saplings are sheltered and not exposed to the atmosphere, and therefore a normal occurrence above treelines also without any warming trend. To directly infer a climate-driven upward shift of treeline position as hypothesized in Fig. 1, you would need to compare actual treelines. An upward shift of saplings as presented in Fig. 2a is a potential indication at best, especially given the very small increase of 5 m (!), which is well below what you would expect from the climate data (0.4°C warming since 1992, so about 50 m using standard lapse rates). The raw number of seedlings (as in Fig. 2c) is no indication at all. However, I fully agree with your conclusion that by increasing sapling mortality (effectively removing any saplings above treeline), fire has the potential to impede the upslope movement of subalpine forests. If only temporarily or permanently depends on future fire frequencies and a potential shift in fire regimes.

- Forest fires normally depend on two factors: climate and fuel. Without enough fuel there is no fire. So, I was wondering what the fuel loads above or at treeline look like and how far fires can penetrate into the alpine vegetation as brush (grass?) fires. Could you comment on this? I agree that since sapling establishment is restricted to areas close to the treeline (10 m) fire would still kill of most saplings (as you show) but what about single individuals located farther up? Would they still be affected? And with temperature projections of up to +3.8°C could single individuals establish above the zone affected by forest fires, if seeds somehow get there?

I recommend the article for publication in PLOSone after addressing these remarks and some minor comments below:

L. 158: How are “large adult trees” defined? >2 m height? Basal diameter >25 cm?

L. 237: How was the age of saplings estimated? Just by diameter? Because in Fig. 3 the oldest “saplings” have supposedly established in 1939, which means that they are not technically saplings (as in “young trees”) anymore, but rather quite old trees that were growing very slowly due to the harsh conditions above treeline.

L. 240-241: This is a bit confusing, since in the previous sentence you mention that “…mortality above treeline was high.” And here you state that “Tree mortality was low…”. I guess you talk about actual dead individuals found, whereas in the previous sentence you base your interpretation on the difference between the survey period, which clearly also includes seedlings. I suggest rephrasing to make this distinction clear.

L.297-310: I fully agree, what you observe here, a large number of seedlings/saplings above the treeline with very low to zero long-term survival due to microclimatic conditions (ground vs. athmosphere) is a very common feature of treeline ecosystems around the world. But this is also why it is not a good idea to use seedling/sapling establishment above treelines as an indicator of an upslope movement of treelines in response to warming temperatures.

L. 326: single not singe

Reviewer #2: This is an excellent paper: It is well written, the methods and results are clear, and the authors use the international literature to frame the introduction and provide a nuanced discussion. They build on existing conceptual models, and reframe them for plants with different traits and life history strategies.

The main limitation is that they only have four sites which each have unique grazing and disturbance histories. In otherwords, this study is a comparison of four case studies. While inferences would be stronger with more sites, these case studies are still valuable. The inferences the authors draw from those sites are sound, and are important for predicting how treelines worldwide will shift in response to climate change.

A second limitation is that they are inferring that fire is the main cause of demographic changes in tree populations. This may be very reasonable to do, given the dynamics of these ecosystems, but the authors should add some context for international readers: are how do they know that the fires were the main cause of these changes? Can the rule out windthrow, avalanches, insects, etc? The authors need to add text providing this context.

This study will make an excellent contribution to the global literature on treeline dynamics and wildfire interactions at ecotones.

The authors state the data will be made available but they have not done so yet.

42-64 This is a really nice discussion of resprouting, and the variety of ways resprouting occurs in species and populations. Nice set-up for the whole article.

Figure 1. I think it is great to include a conceptual diagram like this, which allows readers to visualize the dynamics of your study ecosystem. I’m familiar with the other paper you cite, which presents a similar model for obligate-seeding conifer forests. This is a nice modification for resprouting forests. Fig. 1d, is probably the most interesting, with the interaction with climatic stress and vulnerability to repeat fire. You don’t show it, but I can think of treelines where vulnerability to repeat fire may be greatest further away from the climatic limit to growth. In fact, I think the original paper you cite suggests that is the case in North America temperate forests. You might want to make that clear, and consider again (more explicitly) in the discussion how interactions between stress and fire impact immediate ecotone patterns, and long-term change.

132: S1 Table 1 shows grazing history as well, which varied greatly between some of the sites. Do you need to mention that history in the text?

132: S1 Fig1: This is an excellent map. I’d suggest including it in the main manuscript. This context is needed for any international readers.

149-154: Given the current fires in NSW, you many need to modify lines 149-151 to say “Since 2000 to the time of measurement in 2018…” And then you may need to add a line to note that the pattern of repeat burning is continuing. It looks like “The Twins” site is within the boundary of active MODIS returns of the current fires.

(The extent if the current wildfire impacts in NSW and Victoria are dominating the news worldwide right now. Simply suggesting edits to your text without acknowledging the scale of the ongoing human disaster seemed inappropriate. My condolences if the current fires are impacting people you know, in addition to your research sites.)

148-149: Disturbance rates should be described using standard frequency measures in fire ecology, such as fire rotation or mean fire return interval, if possible.

157-165: Methods are clear and defensible. Good.

169-170: If neither of the two permanent pegs at a transect could not be found, was a transect dropped? or did you re-establish it using GPS coordinates. Either may be fine (depending on the original GPS accuracy), but re-write this sentence to make your re-measurement methods clear.

172-177: Were your field protocols simply confirming—at the tree or transect scale—what was already shown on maps of fire perimeters? If so, make this clear.

139: Condit et al.’s method was developed to tropical trees, but has been wildly applied. You use of it seems reasonable given your goals.

271: You could also cite Wang et al. 2019 here – you cite it elsewhere.

301-306: You may want to look at and possibly cite this recent experimental paper on freezing damage. It is from North America, not Australia, but well-controlled experiments like this one are rare.

Maher, C.T., Nelson, C.R. and Larson, A.J., 2019. Winter damage is more important than summer temperature for maintaining the krummholz growth form above alpine treeline. Journal of Ecology.

312-322: Good discussion of the lack of upslope migration. This fits well with Harsch et al’s conceptual paper on “treeline forms”—and their description of the limiting factors for “abrupt treelines”—which you cite in the conclusion. I think it would be appropriate to cite here, and perhaps draw in that conceptual framework to this discussion.

344-346: These two lines repeat verbatim information presented in the introduction on lines 122-124. Re-phrase.

Figures: In the .pdf I was provided, it is very hard to read the axis labels and tic labels of the figures, particularly for Figure 2. I was able to make sense of it from the captions, though. The figures may be published in higher resolution, but the font size should also be increased.

Figure 3: Could you add vertical dashed lines to this figure, showing the time of disturbance(s) at each site? That type of visual reference would be helpful.

Figure 4: Consider adding the summary statistics as text to each panel in this figure: the data, slope, and what type it is. Then define the types in the figure caption. This would allow the figure to stand alone, and not be reliant on the text.

6. PLOS authors have the option to publish the peer review history of their article (what does this mean?). If published, this will include your full peer review and any attached files.

Reviewer #1: No

Reviewer #2: No

---

## [Author Response · Author response to Decision Letter 0]

4 Mar 2020

Thank you for revising our manuscript and please find below the changes to the Data Availability Statement, Funding Statement and Copyright information for Figure S1 as per your email.

Changes to Data Availability Statement:

We would like to change our Data Availability statement to “The data that supports the findings of this study are openly available at Harvard Dataverse at https://doi.org/10.7910/DVN/I5MNND. 

Changes to Funding Statement:

We would like to change our Funding statement to “The Research Centre for Applied Alpine Ecology provided financial support”. 

Figure S1 (Revised to Figure 2) Copyright:

We sought copyright permission from the Department of Environment, Land, Water and Planning, through the Spatial Datamart Victoria. The Department of Environment, Land, Water and Planning replied that the metadata is licenced under the Creative Commons Attribution 4.0 International licence, Copyright and Attribution, and thus we are free to use the dataset as per the licence and there is no need for copyright permission. The spatial data was sourced from the Spatial Datamart Victoria: https://services.land.vic.gov.au/SpatialDatamart/dataSearchViewMetadata.html?anzlicId=ANZVI0803004741&extractionProviderId=1

This link includes the copyright information under “Access constraints”. 

For the response to reviewers comments please see attached 'Response to Reviewers' document.

---

## [Editor Report · Decision Letter 1]

23 Mar 2020

Alpine treeline ecotone stasis in the face of recent climate change and disturbance by fire

PONE-D-19-29648R1

Dear Dr. Naccarella,

We are pleased to inform you that your manuscript has been judged scientifically suitable for publication and will be formally accepted for publication once it complies with all outstanding technical requirements.

With kind regards,

Sergio Rossi

Academic Editor

PLOS ONE
---

## [Editor Report · Acceptance letter]

25 Mar 2020

PONE-D-19-29648R1 

Alpine treeline ecotone stasis in the face of recent climate change and disturbance by fire 

Dear Dr. Naccarella:

I am pleased to inform you that your manuscript has been deemed suitable for publication in PLOS ONE. Congratulations! Your manuscript is now with our production department. 

With kind regards,

on behalf of

Prof. Sergio Rossi 

Academic Editor

PLOS ONE